# The relationships between obesity and epilepsy: A systematic review with meta-analysis

**Yu-xuan Li**[1]⊕, **Wang Guo**[1]⊕, **Ruo-xia Chen**[1‡], **Xue-rui Lv**[1‡], **Yun Li**[2]*

**1** Clinical Medical School, Dali University, Dali, China, **2** Department of Neurology, The First Affiliated Hospital of Dali University, Dali, China

⊕ These authors contributed equally to this work.
‡ RC and XL also contributed equally to this work.
* yy_neuron@163.com

**Data Availability Statement:** All relevant data are within the manuscript and its Supporting Information files.

**Funding:** The author(s) received no specific funding for this work.

## Abstract

### Objective

There is ongoing debate regarding the association between epilepsy and obesity. Thus, the aim of this study was to examine the correlation between epilepsy and obesity.

### Method

This study adhered to the PRISMA guidelines for systematic reviews and meta-analyses. On The Prospero website, this study has been successfully registered (CRD42023439530), searching electronic databases from the Cochr-ane Library, PubMed, Web of Sciences and Embase until February 10, 2024.The search keywords included "Epilepsy", "Obesity", "Case-Control Studies", "cohort studies", "Randomized Controlled Trial" and "Cross-Sectional Studies". The medical subject headings(MeSH) of PubMed was utilized to search for relevant subject words and free words, and a comprehensive search strategy was developed. Two reviewers conducted article screening, data extraction and bias risk assessment in strict accordance with the predefined criteria for including and excluding studies. The predefined inclusion criteria were as follows: 1) Inclusion of case-control, cohort, randomized controlled trial, and cross-sectional studies; 2) Segregation of subjects into epileptic patients and healthy controls; 3)Obesity as the outcome measure; 4) Availability of comprehensive data; 5) Publication in English. The exclusion criteria were as follows: 1) Exclusion of animal experiments, reviews, and other types of studies; 2) Absence of a healthy control group; 3) Incomplete data; 4) Unextractable or unconvertible data; 5) Low quality, indicated by an Agency for Healthcare Research and Quality(AHRQ) score of 5 or lower,or a Newcastle-Ottawa Scale (NOS) score less than 3. The subjects included in the study included adults and children, and the diagnostic criteria for obesity were used at different ages. In this study, obesity was defined as having a body mass index(BMI) of 25 kg/m² or higher in adults and being above the 85th percentile of BMI for age in children. We used obesity as an outcome measure for meta-analysis using RevMan, version 5.3.

**Competing interests:** The authors have declared that no competing interests exist.

## Results

A meta-analysis was conducted on a total of 17 clinical studies, which involved 5329 patients with epilepsy and 480837 healthy controls. These studies were selected from a pool of 1497 articles obtained from four electronic databases mentioned earlier. Duplicate studies were removed based on the search strategies employed. No significant heterogeneity was observed in the outcome measure of obesity in epileptic patients compared with healthy controls($p = 0.01, I^2 = 49\%$). Therefore, a fixed effects model was utilized in this study. The findings revealed a significant difference in obesity prevalence between patients with epilepsy and healthy controls(OR = 1.28, 95%CI: 1.20–1.38, p<0.01).

## Conclusion

The results of this meta-analysis indicate that epilepsy patients are more prone to obesity than healthy people, so we need to pay attention to the problem of post-epilepsy obesity clinically. Currently, there is a scarcity of largescale prospective studies. Additional clinical investigations are warranted to delve deeper into whether obesity is a comorbidity of epilepsy and whether obesity can potentially trigger epilepsy.

## Background

Epilepsy is a chronic neurological disorder [1]. Epilepsy is defined as at least 2 non-induced (or reflexive) seizures, with an interval of more than 24 hours. The definition also takes into account the risk of non-induced seizures and a diagnosis of epileptic syndrome within the next 10 years [2]. Globally, the burden of neurological diseases is on the rise [3]. Epilepsy, in particular, affects over 70 million individuals worldwide [4] and results in the annual deaths of 125,000 patients [5, 6]. Individuals with epilepsy frequently experience comorbidities such as cognitive impairment, depression, and obesity [7, 8]. Therefore, it is crucial to focus on these comorbidities associated with epilepsy.

Numerous studies have indicated that sodium valproate can induce obesity and elevate the risk of polycystic ovary syndrome [9–11]. This prompts the question: Are individuals with epilepsy more prone to obesity due to factors like antiepileptic drugs and reduced physical activity? The correlation between obesity and epilepsy is still a controversial topic. Zhou K conducted a Mendelian randomization study and found that obesity and epilepsy are causally related to each other [12]. Additionally, researchers like Buro AW, Lee DH, Lee Y, Arya R et al have argued that individuals with epilepsy are more susceptible to obesity [13–16]. Chen M, Nazish S et al have posited a positive correlation between overweight and epilepsy. Conversely, some studies suggest no association between obesity and epilepsy [17, 18]. Pfeifer MT's cross-sectional study even found a significant positive correlation between underweight and epilepsy [19]. Meanwhile, Janousek J, Adane T, Inaloo S et al. contended that the obesity rate among epilepsy patients resembles that of the general population [20–22].

In conclusion, the correlation between epilepsy and obesity remains contentious, lacking a definitive consensus regarding their relationship. While some studies have linked obesity to epilepsy, and others have associated underweight with epilepsy, certain investigations have failed to find any significant correlation. Hence, we conducted an extensive meta-analysis to delve deeper into this association and examine potential influencing factors. Our aim is to offer valuable theoretical insights into the pathogenesis of epilepsy. This research is significant

in preventing and managing epilepsy, laying the groundwork for personalized intervention strategies. It provides valuable guidance for managing body mass index in epilepsy patients and optimizing tailored treatment approaches. Additionally, it aids in identifying potential therapeutic targets for future exploration.

## Methods

### Search strategy

This study followed the PRISMA guidelines for systematic review and meta-analysis. It was registered with Prospero (CRD42023439530) and involved an extensive search of electronic databases, including the Cochrane Library, PubMed, Web of Science, and Embase, up to February 10,2024. The search incorporated keywords such as "Epilepsy", "Obesity", "Case-Control Studies", "cohort studies", "Randomized Controlled Trial" and "Cross-Sectional Studies". Develop comprehensive search strategies with PubMed MeSH.

### Research eligibility

Two independent reviewers conducted the assessment of study eligibility, data extraction, and evaluation of bias risk. Quality assessment was conducted using the Newcastle-Ottawa Scale (NOS) for case-control studies and the Agency for Healthcare Research and Quality (AHRQ) for cross-sectional studies. The predefined inclusion criteria were as follows: 1) Inclusion of case-control, cohort, randomized controlled trial, and cross-sectional studies; 2) Segregation of subjects into epileptic patients and healthy controls; 3) Obesity as the outcome measure; 4) Availability of comprehensive data; 5) Publication in English. The exclusion criteria were as follows: 1) Exclusion of animal experiments, reviews, and other types of studies; 2) Absence of a healthy control group; 3) Incomplete data; 4) Unextractable or unconvertible data; 5) Low quality, indicated by an Agency for Healthcare Research and Quality(AHRQ) score of 5 or lower, or a Newcastle-Ottawa Scale (NOS) score less than 3. This study categorized adult obesity into four groups based on body mass index(BMI): underweight (BMI<18.5kg/m$^2$), normal weight (BMI 18.5–24.9kg/m$^2$), overweight(BMI 25–29.9kg/m$^2$), and obesity(BMI $\geq$ 30kg/m$^2$) [19]. The BMI percentiles of children were divided into the following categories: 1) obese: BMI $\geq$ 95th percentile for age, 2) overweight: 85th percentile $\leq$ BMI < 95th percentile for age, 3) Healthy weight: 10th percentile $\leq$ BMI < 85th percentile for age, and 4) Underweight: BMI < 10th percentile for age [23]. In this study, obesity was defined as a body mass index (BMI) equal to or greater than 25kg/m$^2$ for adults and above the 85th percentile of BMI for children based on age.

### Data extraction

We extracted data from the included papers, including the first author and publication year, country, journal, groupings, sample sizes, age, sex, influence factors and the primary outcome (e.g., BMI).

### Data analysis

We utilized the incidence of obesity as a binary outcome measure and conducted a meta-analysis using RevMan, version 5.3. We employed the mean difference as the effect measure, presenting each effect size with a corresponding 95% confidence interval(CI). To assess heterogeneity, we employed the Q test and calculated the I$^2$ statistic. In cases where p>0.05 and I$^2$<50%, we regarded heterogeneity as nonsignificant and opted for the fixed-effect model. When the pvalue was less than 0.05 and the I$^2$ statistic exceeded 50%, subgroup analysis

was performed to identify the source of heterogeneity. If heterogeneity persisted, a random-effects model was utilized.

## Results

### Literature search results

Initially, 1497 articles were retrieved from four electronic databases using our search strategy, resulting in 313 articles after removing duplicates. Upon title and abstract evaluation, 837 irrelevant articles were excluded, leaving 214 articles for further examination. After full-text review, 100 studies lacked healthy controls, 4 articles were inaccessible in full, 49 studies had unextractable or untranslatable data, and 43 were unrelated studies, all of which were excluded. In total, 17 studies were included, comprising 13 of high quality and 4 of medium quality. The review process is depicted in Fig 1. Table 1 details the eligible studies. In the included articles, we identified various factors influencing epilepsy and obesity, such as anti-epileptic medications, reduced physical activity, dietary habits, age, gender, ethnicity, and psychiatric disorders (e.g., anxiety, depression). Please refer to Table 2 for details.

### Meta-analysis results

The analysis encompassed 17 clinical studies involving 5329 patients with epilepsy and 480837 healthy controls. Our initial assessment did not reveal significant heterogeneity in the outcome indicators for obesity in epileptic patients compared to healthy controls (p = 0.01, $I^2$ = 49%) (Fig 2). As a result, we conducted a meta-analysis utilizing a fixed-effect model. The study

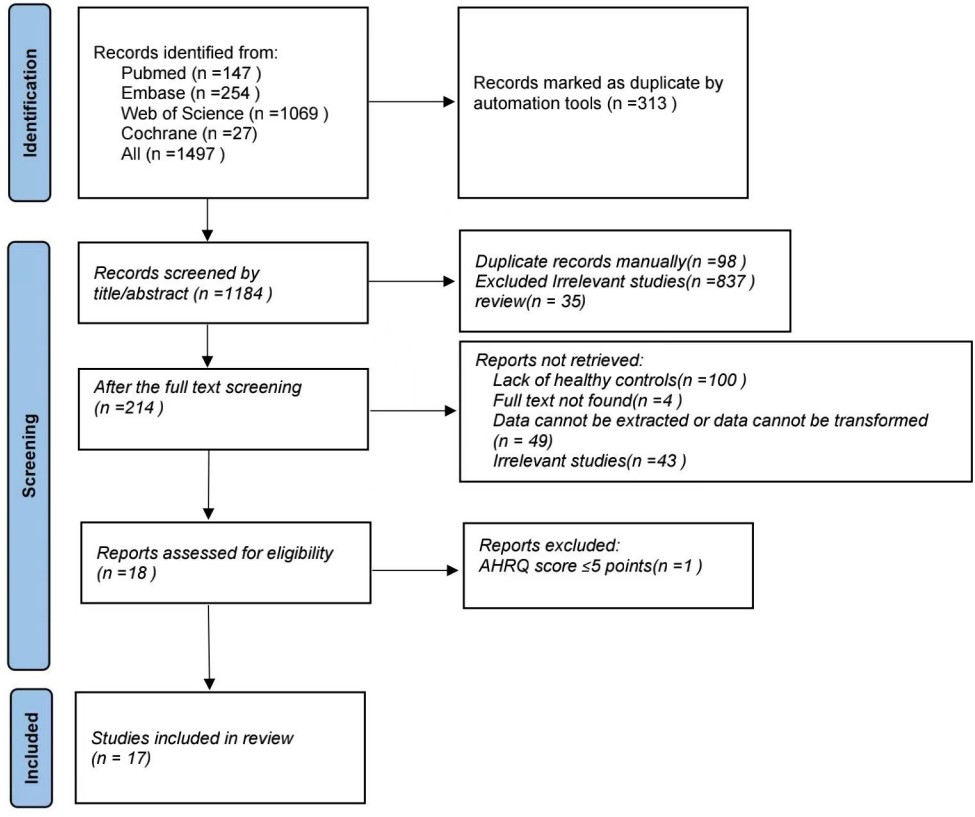

**Fig 1. Flowchart illustrating the study selection process for inclusion in the meta-analysis.**

**Table 1. Basic features of the included literature.**

| First Author, Year | Country | Journal | Study Type | Group(n) | Female/ Male (n) | Age(Years/'X ±S) | Obesity (n) | Quality Score (AHRQ /NOS) |
|---|---|---|---|---|---|---|---|---|
| Pylvänen, V, 2002 [24] | Finland | Epilepsia | Case-Control Studies | Epileptic 81 | 35/46 | 29.6 ± 10.9 | 40 | 7[a] |
| | | | | healthy control group 51 | 27/24 | 37.1±7.1 | 25 | |
| Pylvänen, V, 2003 [25] | Finland | Neurology | Case-Control Studies | men with epilepsy 102 | 0/102 | 32–45.6 | 50 | 7[a] |
| | | | | healthy men 32 | 0/32 | | 20 | |
| Marquez AV, 2003 [26] | USA | Epilepsy Behav | Cross-Sectional Studies | Epileptic46 | —— | —— | 21 | 7[b] |
| | | | | Healthy control group46 | | | 29 | |
| El-Khayat, H. A, 2004 [27] | Egypt | Epilepsia | Case-Control Studies | Women with epilepsy 66 | 66/0 | 8–18 | 18 | 8[a] |
| | | | | healthy women 40 | 40/0 | | 4 | |
| Kobau R, 2004 [28] | USA | Epilepsy Behav | Cross-Sectional Studies | Epileptic161 | 100/61 | —— | 107 | 8[b] |
| | | | | healthy control group7896 | —— | | 4533 | |
| Pylvänen, V, 2006 [29] | Finland | Epilepsia | Case-Control Studies | Epileptic 51 | 20/31 | 31.4 ± 11.9 | 27 | 6[a] |
| | | | | Healthy control group45 | 22/23 | 30.9±8.5 | 24 | |
| Wong J, 2006 [30] | Canada | Epilepsia | Cross-Sectional Studies | Epileptic79 | —— | 5–17 | 18 | 9[b] |
| | | | | Non-Epileptic99 | | | 17 | |
| Elliott JO, 2008 [31] | USA | Epilepsy Behav | Cross-Sectional Studies | Epileptic604 | 393/211 | >18 | 346 | 7[b] |
| | | | | healthy control group42416 | 25155/ 17261 | | 23302 | |
| Z. S. Daniels, BA, 2009 [23] | USA | Neurology | Case-Control Studies | Epileptic 251 | 119/132 | 8.5–14.7 | 97 | 8[a] |
| | | | | healthy control group 597 | 301/296 | 5.4–12 | 170 | |
| Hinnell, C, 2010 [32] | Canada | Epilepsia | Cross-Sectional Studies | Epileptic 2555 | —— | 43 ± 17.7 | 488 | 8[b] |
| | | | | Healthy control group 400055 | —— | 45.4± 20.2 | 61608 | |
| Mania M, 2011 [33] | Georgia | Georgian Med News | Case-Control Studies | Epileptic 54 | 27/27 | 33±11 | 15 | 7[a] |
| | | | | Healthy control group25 | —— | 32±10 | 3 | |
| Ayyagari, M, 2012 [34] | India | Ann Indian Acad Neurol | Case-Control Studies | Women with epilepsy 60 | 60/0 | 13–45 | 12 | 7[a] |
| | | | | healthy women 20 | 20/0 | | 5 | |
| Arya, R, 2016 [16] | USA | Neurology | Cross-Sectional Studies | Epileptic 445 | —— | 2.5–13 | 151 | 9[b] |
| | | | | healthy control group2079 | —— | | 526 | |
| Inaloo S, 2020 [22] | Iran | Iran J Child Neurol | Case-Control Studies | Epileptic 77 | 38/39 | 11.4 ± 3.2 | 5 | 9[a] |
| | | | | Healthy control group77 | 38/39 | | 0 | |
| Khuda, I. E, 2022 [35] | Saudi Arabia | Prim Care Companion CNS Disord | Cross-Sectional Studies | Epileptic 110 | 62/48 | >20 | 54 | 8[b] |
| | | | | healthy control group 46 | 26/20 | | 14 | |
| Tadegew Adane, 2023 [21] | Ethiopia | Neuropsychia-tric Disease and Treatment | Cross-Sectional Studies | Epileptic 403 | 224/179 | 28.32±5.92 | 30 | 8[b] |
| | | | | Non-Epileptic 403 | 189/214 | 25.47±6.09 | 38 | |

(*Continued*)

**Table 1.** (Continued)

| First Author, Year | Country | Journal | Study Type | Group(n) | Female/ Male (n) | Age(Years/'X ±S) | Obesity (n) | Quality Score (AHRQ /NOS) |
|---|---|---|---|---|---|---|---|---|
| Buro AW, 2023 [36] | USA | J Child Neurol | Cross-Sectional Studies | Epileptic184 | 86/98 | 10–17 | 33 | 11[b] |
| | | | | Non-Epileptic 26910 | 12973/ 13973 | | 3394 | |

Remark: Quality evaluation: [a]Score for NOS quality evaluation form
[b]Score for AHRQ quality evaluation form

found a significant difference in obesity rates between individuals with epilepsy and healthy controls(OR = 1.28, 95%CI: 1.20–1.38, p<0.01). To assess publication bias, we employed funnel plots(Fig 3). Sensitivity analysis was performed to observe and look for possible heterogeneity in this study, involving the substitution of the fixed-effects model with a random-effects model (Fig 4), revealing no significant heterogeneity(p = 0.01, I2 = 49%). Additionally, we conducted sensitivity analysis by excluding studies with the highest weight(Fig 5), which did reveal heterogeneity(p<0.05, $I^2$ = 52%), suggesting instability in the meta-analysis results.

## Subgroup analysis

1. Subgroup analysis was based on the age of the subjects(Fig 6).

**Table 2. Factors affecting epilepsy and obesity.**

| First author,Year | influence factor |
|---|---|
| Pylvänen, V, 2002 [24] | VPA |
| Pylvänen, V, 2003 [25] | VPA,CBZ,OXC |
| Marquez AV, 2003 [26] | psychiatric illness |
| El-Khayat, H. A, 2004 [27] | VPA |
| Kobau R, 2004 [28] | reductions in exercise,increase eating (e.g.,VPA,GABA),increased lipogenesis (as suggested for VPA),several older AEDS (e.g.,PB) |
| Pylvänen, V, 2006 [29] | VPA |
| Wong J, 2006 [30] | Lack of physical activity |
| Elliott JO, 2008 [31] | AEDs,increased rates of depression,low income,exercise and diet behaviors,regional differences |
| Z. S. Daniels, BA, 2009 [23] | increasing age,idiopathic etiology,AEDs(i.e.,VPA,GABA) |
| Hinnell, C, 2010 [32] | less physically active,AEDs,psychiatric comorbidities |
| Mania M, 2011 [33] | VPA |
| Ayyagari, M, 2012 [34] | CBZ, VAP, PHT |
| Arya, R, 2016 [16] | Lack of physical activity,VPA,LTG |
| Inaloo S, 2020 [22] | VPA,a sedentary lifestyle |
| Khuda, I. E, 2022 [35] | a lack of physical activity,inappropriate dietary habits |
| Tadegew Adane, 2023 [21] | a lack of physical activity |
| Buro AW, 2023 [36] | age, sex, race/ethnicity,mental health,family functioning,built environment,forgone care |

Remark: Anti-epileptic drugs (AEDS), Valproate (VPA), Carbamazepine (CBZ), Oxcarbazepine (OXC), Phenytoin (PHT), Phenobarbital(PB), Ethosuximide(ESM), Gabapentin(GABA), Lamotrigine(LTG)

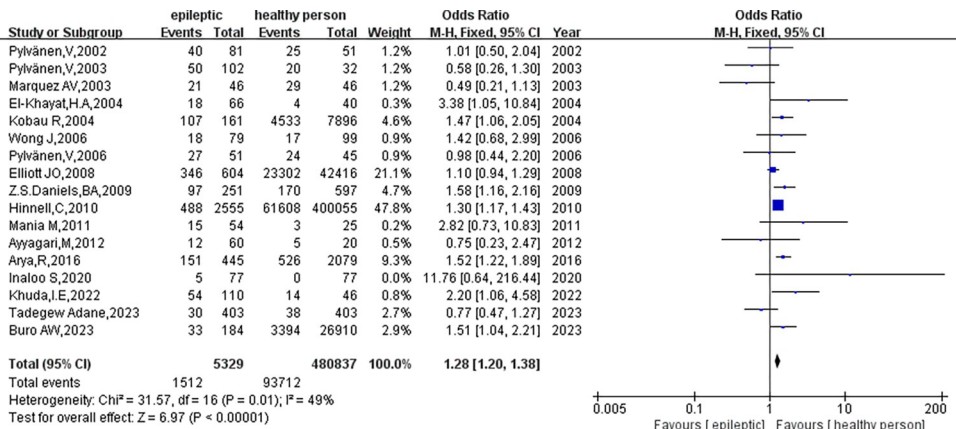

**Fig 2. A meta-analysis of obesity in epileptic and healthy controls.**

① Studies involving adults did not find significant heterogeneity(p = 0.05, $I^2$ = 45%) and showed statistically significant differences in obesity(OR = 1.24, 95%CI: 1.15–1.34, p<0.01). ② Conversely, there was significant heterogeneity in the results of the study involving children(p = 0.06, $I^2$ = 53%), with the results demonstrating a statistically significant difference in obesity(OR = 1.47, 95%CI: 1.25–1.72, p<0.01). The subgroup difference test(p = 0.07, $I^2$ = 69.1%) indicated that Age was associated with the heterogeneity observed in this meta-analysis.

2. Subgroup analysis based on the year of publication (Fig 7).

① There was no significant heterogeneity in the results of articles published within the last 10 years (p = 0.08, $I^2$ = 49%), and the results showed a significant difference in obesity between epilepsy patients and healthy subjects (OR = 1.48, 95%CI: 1.27–1.72, p<0.01). Results

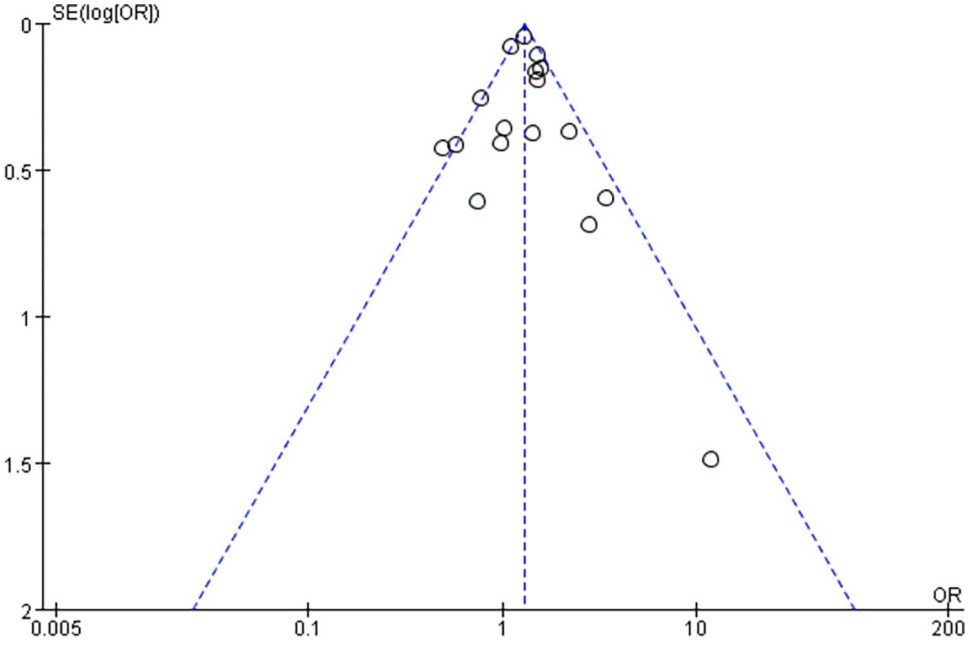

**Fig 3. Funnel plots for all studies.**

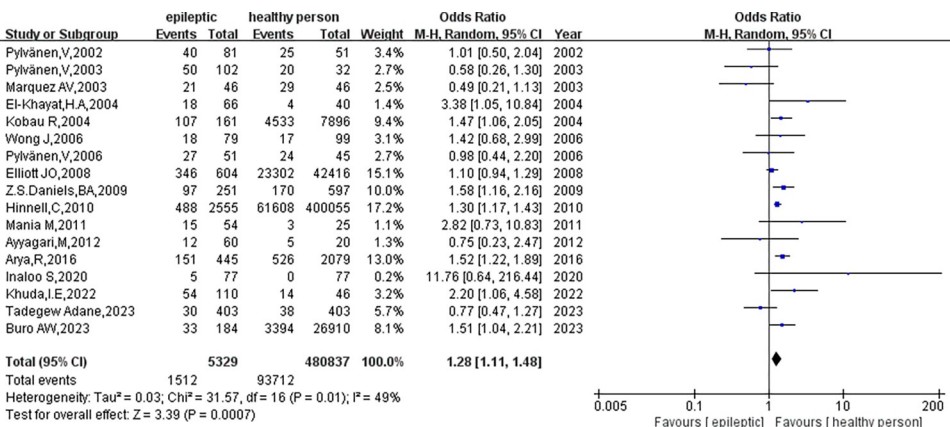

**Fig 4. The fixed effects model is replaced by a random effects model for sensitivity analysis.**

published more than 10 years ago did not find significant heterogeneity (p = 0.06, $I^2$ = 44%), but showed a statistically significant difference in obesity (OR = 1.23, 95%CI: 1.14–1.34, p<0.01). Subgroup difference tests (p = 0.04, $I^2$ = 76.7%) showed that the year of publication was associated with the observed heterogeneity.

3. Subgroup analysis based on economic development level (Fig 8)

① In the developed countries, there was significant heterogeneity (p = 0.02, $I^2$ = 54%), and the difference in obesity was statistically significant (OR = 1.26, 95%CI: 1.17–1.35, p<0.01). In contrast, studies from developing countries found no significant heterogeneity (p = 0.28, $I^2$ = 21%) and showed statistically significant differences in obesity (OR = 1.81, 95%CI: 1.35–2.42, p<0.01). A subgroup difference test (p = 0.02, $I^2$ = 81.7%) showed that the level of published economic development was associated with the heterogeneity observed in this study.

In summary, this study detected heterogeneity through sensitivity analysis by excluding studies with the highest weight, highlighting the need for caution in interpreting the results due to potential instability. To explore the sources of heterogeneity, subgroup analysis was conducted, revealing age, publication years of included articles, and the level of economic development as potential factors. The study encompassed both adults and children, acknowledging variations in the prevalence of post-epileptic obesity across different age groups, which could contribute to heterogeneity among studies. Nonetheless, subgroup analysis still indicates

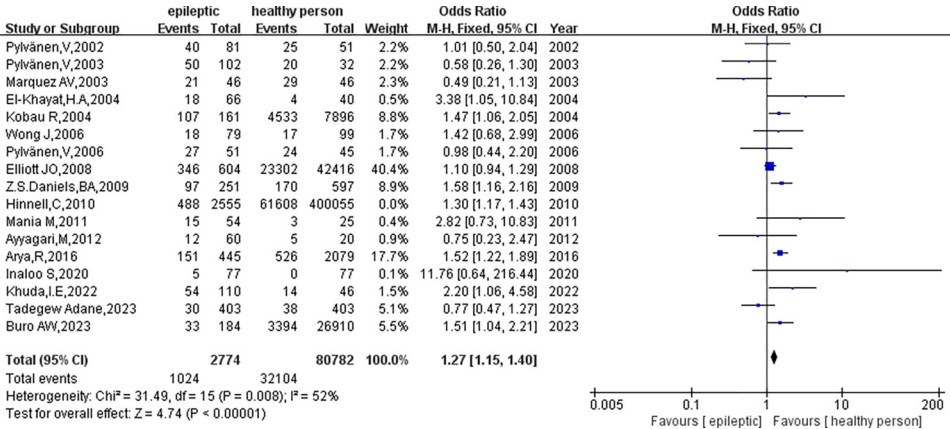

**Fig 5. The most weighted studies were excluded for sensitivity analysis.**

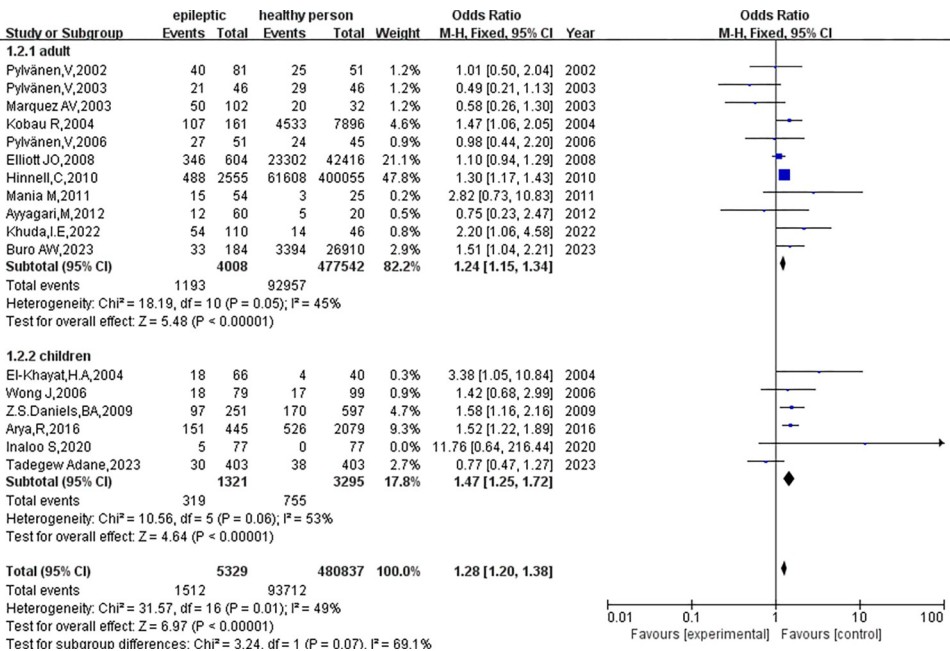

**Fig 6. Subgroup analysis was performed according to the age of the subjects.**

a correlation between epilepsy and obesity in both adults and children. Additionally, considering that the majority of included studies were published a decade ago and primarily conducted in developed countries, there is a possibility of time bias in the results, while economic development could also influence obesity outcomes. Subgroup analysis further highlighted that publication years of articles and economic development at research sites contributed to the observed heterogeneity.

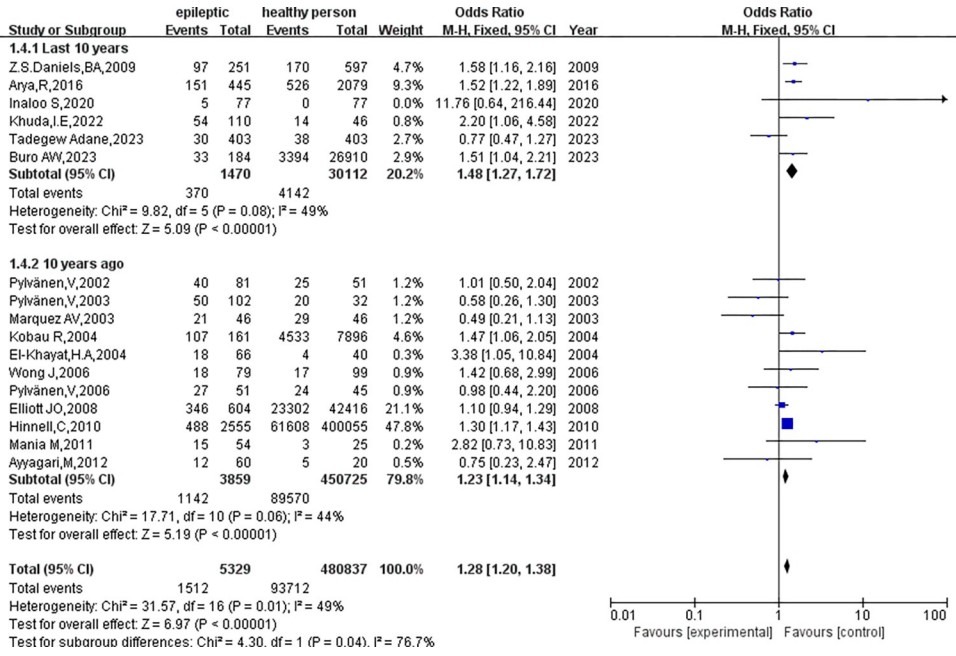

**Fig 7. Subgroup analysis based on years of publication.**

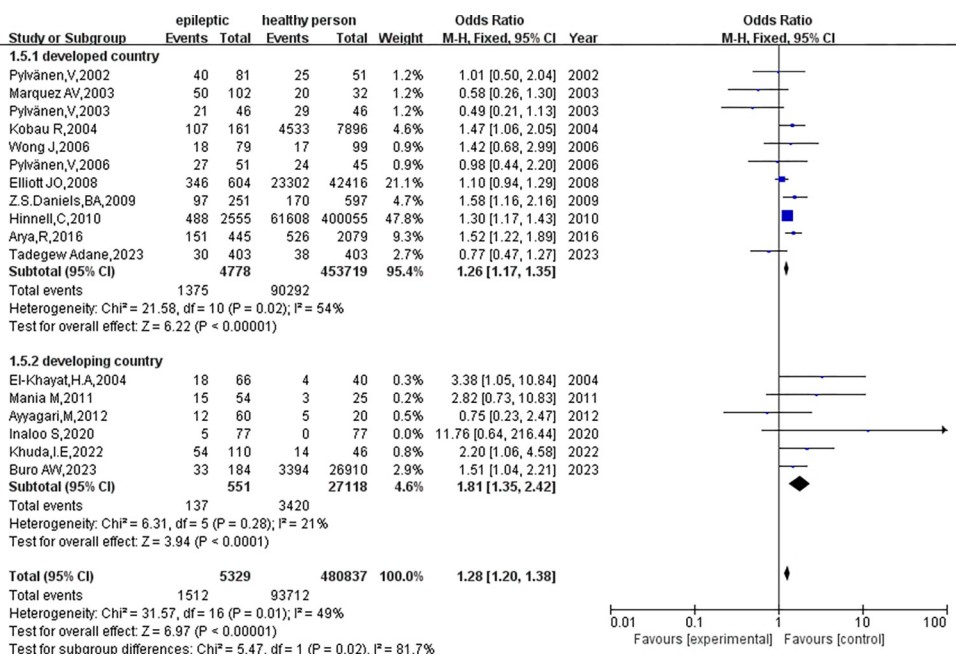

**Fig 8. Subgroup analysis was conducted according to the level of economic development.**

## Discussion

There is an increasing focus on improving the well-being of individuals with epilepsy and gaining a better understanding of the additional health issues that often accompany this condition. Research has identified that epilepsy patients are susceptible to cognitive disorders, depression, obesity, and endocrine disturbances. However, the link between epilepsy and obesity remains a subject of debate, and the underlying mechanisms connecting these two conditions remain unclear. The results of this meta-analysis suggest that individuals with epilepsy have a higher likelihood of being obese compared to those without the condition. Consequently, it raises the question: are obese individuals more prone to developing epilepsy than their healthy counterparts, or do individuals with epilepsy have a higher likelihood of becoming obese? It's important to note that this meta-analysis comprises case-control studies, limiting the ability to draw causal inferences. Zhou K's work, utilizing reverse Mendelian randomization and multiple Mendelian randomization, discovered a bidirectional causal relationship between epilepsy and obesity in adolescents [11]. However, whether this same causal relationship exists in adults

**Table 3. Effects of antiepileptic drugs on obesity.**

| Sort | Gain Weight | Lose Weight |
| --- | --- | --- |
| The first generation of AEDS | VPA, CBZ, PHT, CLB, ESM, PB [47–49] | / |
| Second generation AEDS | OXC [50], GABA, PGB [51] | TPM [42–44], LTG [45] |
| Third generation AEDS | LCM [52], PER [53] | ZNS [46], FENF [44] |

Remark: Anti-epileptic, drugs(AEDS), Carbamazepine(CBZ), Phenobarbital(PB), Phenytoin(PHT), Valproate(VPA), Ethosuximide(ESM), Clobazam(CLB), Gabapentin(GABA), Lamotrigine(LTG), Levetiracetam(LEV), Oxcarbazepine (OXC), Pregabalin(PGB), Topiramate(TPM), Zonisamide(ZNS), Lacosamine(LCM), Pirampanil(PER), Fenfluramine (FENF)

remains unconfirmed by Mendelian studies. Thus, the precise mechanisms underpinning the connection between obesity and epilepsy remain uncertain and warrant further investigation.

Currently, the predominant focus of research lies in examining the impact of antiepileptic drugs on obesity, with particular attention to sodium valproate. Sodium valproate is a commonly prescribed medication for various types of seizures, including simple or complex absence seizures, myoclonic seizures, and severe seizures, either alone or in combination. Its cost-effectiveness has contributed to its widespread clinical usage. Numerous studies have established that sodium valproate is associated with several adverse effects, including elevated levels of leptin, insulin resistance, increased leptin/adiponectin ratio, hyperinsulinemia, elevated body mass index (BMI), increased blood lipid levels, reduced carnitine, significant reductions in bone mineral density, elevated liver enzyme levels, menstrual irregularities, altered reproductive hormone function, mitochondrial dysfunction, endocrine disturbances, and women of childbearing age are more likely to develop PCOS [9, 10, 37–40]. Furthermore, some studies have suggested that sodium valproate-induced obesity and weight gain might be linked to CD36 and PPARγ polymorphisms, with these genetic factors potentially serving as predictive markers for sodium valproate-induced obesity in Chinese Han epilepsy patients [41]. However, research concerning the effects of other antiepileptic drugs on obesity remains limited and contentious. Existing evidence suggests that certain medications such as Topiramate(TPM) [42–44], Lamotrigine(LTG) [45], Zonisamide(ZNS) [46] and Fenfluramine (FENF) [44] may contribute to weight reduction [42–44], whereas Valproate(VPA), Carbamazepine(CBZ), Phenytoin(PHT), Clobazam(CLB), Ethosuximide(ESM), Phenobarbital(PB) [47–49], Oxcarbazepine(OXC) [50], Gabapentin(GABA), Pregabalin(PGB) [51], Lacosamine (LCM) [52] and Pirampanil(PER) [53] tend to promote weight gain [47–53]. The effects of different antiepileptic drugs on obesity are shown in Table 3. Horizontal genetic studies have found that SLC13A5 epilepsy is caused by newborn SLC13A5 mutations. The sodium-dependent citric acid transporter NaCT, which is encoded by SLC13A5, could be a promising target for interventions aimed at combating obesity [54]. It is conceivable that future research may unveil more genetic connections between epilepsy and obesity, potentially opening new avenues for the treatment of epilepsy. In clinical practice, it is essential to monitor the impact of various antiepileptic drugs on blood lipid profiles and body weight while also considering the cardiovascular and cerebrovascular risks associated with these medications.

The relationship between epilepsy and obesity is complex. While there is no conclusive evidence that epilepsy directly causes obesity, several contributing factors have been identified. Research suggests that poor physical fitness and lower parental education levels are associated with childhood obesity in epilepsy patients [55]. The overall quality of life for epilepsy patients can decline, potentially leading to reduced physical activity and an increased risk of obesity [55]. Yang J emphasizes the need to monitor the BMI of youngsters with epilepsy, particularly those from less educated and low-income families [55]. Razaz Nalso underscores the importance of preventing obesity as a public health program to Reduce the prevalence rate of epilepsy in children [56]. Therefore, it is crucial to consider epilepsy and its related factors when addressing obesity in clinical practice, especially in the case of pediatric patients. The ketogenic diet has emerged as a promising area of research for the management of epilepsy and obesity. This diet, characterized by high fat, adequate protein, and minimal carbohydrates, induces a state similar to fasting, promoting the production of ketones [57]. Notably, the ketogenic diet has shown potential not only in controlling refractory epilepsy but also in aiding weight loss. Clinical and basic studies suggest that the therapeutic effects of the ketogenic diet may be attributed to its impact on neuronal metabolism, neurotransmitter function, neuronal membrane potential, and neuronal protection [57]. While the ketogenic diet is widely acknowledged as an effective therapy for difficult-to-treat epilepsy, its use in clinical practice remains

limited due to concerns such as poor patient compliance and potential side effects like acute pancreatitis, adverse effects on bone health, elevated monocyte counts, and reduced IgA concentrations [58–60]. However, emerging research indicates that the ketogenic diet may hold therapeutic promise for various diseases beyond epilepsy. Currently, most studies on the ketogenic diet focus on infants and children, with limited research involving adults. Further exploration of the ketogenic diet's effects on adult epilepsy is warranted to optimize treatment strategies. In recent years, some studies have suggested that metformin, a medication commonly used to manage diabetes, may be beneficial in controlling epileptic seizures [61–63]. These findings highlight the importance of monitoring and managing metabolic syndrome and diabetes in epilepsy patients. The potential benefits of implementing a diabetic diet-assisted antiepileptic drug therapy for epilepsy patients also warrant further investigation.

The detrimental effects of obesity on the brain have garnered growing clinical attention [64]. Currently, while it is acknowledged that some anti-epileptic drugs can lead to obesity, there is limited research exploring whether obese individuals are more susceptible to epilepsy. The pathological mechanism underlying obesity-induced epilepsy remains unclear. However, existing research suggests potential connections to various mechanisms. Obesity has been associated with alterations in brain morphology and function, influencing neuroendocrine and neuronal pathways, which in turn impact brain biochemistry and function. Neuroimaging research has shown that obesity is associated with focal structural changes in various brain regions. Some investigations have even established a connection between excess weight and hippocampal volume reduction 17 [65, 66]. In the context of brain function, studies have demonstrated that obesity can trigger an early neuroinflammatory response. Obesity elevates oxidative stress through nicotinamide adenine dinucleotide phosphate oxidase, augments prostaglandin E2 levels, and activates the B-cell nuclear factor-κB light chain enhancer [67, 68]. This cascade results in heightened secretion of pro-inflammatory cytokines and inflammatory markers. This includes interleukin 1, interleukin 6, interleukin 8, interleukin 18, tumor necrosis factor, and C-reactive protein [69, 70], ultimately contributing to metabolic disturbances within the hypothalamus and hippocampus [64]. Furthermore, persistent proinflammatory conditions can impact mitochondrial function and compromise cellular integrity in hypothalamic neurons. This, in turn, results in neuroinflammation, vasculogenes is, alterations in blood-brain barrier integrity, reduced gray matter volume, impaired central glucose perception, and cerebral insulin resistance [64]. Evidence suggests that inflammatory pathways are also implicated in epilepsy's pathophysiology. Chronic activation of microglia in the encephalitic pathway plays a crucial role in both the onset and worsening of seizures, leading to impaired neuronal survival due to increased production of pro-inflammatory cytokines. Inflammatory mediators, such as cyclooxygenase-2 and nuclear factor κB (NF-κB), along with inflammatory factors like interleukin-1β, IL-6, tumor necrosis factor (TNF) -α, and prostaglandin E2 (PGE2), play a significant role in the occurrence and progression of epilepsy [71, 72]. In summary, obesity can induce neuroinflammation, and given the close link between neuroinflammatory pathways and seizures, it is reasonable to infer that obesity may cause epilepsy.

In conclusion, there is compelling evidence to suggest a bidirectional relationship between epilepsy and obesity. Epilepsy patients exhibit a higher susceptibility to obesity compared to the general population, while individuals with obesity may have an increased risk of developing epilepsy. Multiple factors, such as dyslipidemia and inflammatory markers, are likely involved in mediating this association. Therefore, it is imperative for further research to investigate the intricate interplay between these conditions and their underlying influencing factors.

## Limitation

This study is subject to several limitations. Firstly, the predominance of case-control and cross-sectional studies, with limited inclusion of cohort studies and randomized controlled trials, impedes the comprehensive establishment of a causal relationship between epilepsy and obesity. Secondly, heterogeneity among studies may arise due to variations in cultural backgrounds, regions, and measurement tools for outcome indicators, potentially impacting research findings. Additionally, factors such as anti-epileptic drugs, dosage regimens, patient activity levels, disease duration, and seizure frequency mentioned in the literature also exert influence on obesity; however, insufficient data hindered meta-analysis. Large-scale prospective studies are warranted to further investigate whether obesity acts as a comorbidity of epilepsy or if it may serve as a trigger for epilepsy onset. Future endeavors should prioritize rational BMI management and individualized medication strategies for patients with epilepsy. These findings hold significant value in highlighting cardiovascular risks among patients while preventing the occurrence and progression of epilepsy while identifying novel therapeutic targets.

## Supporting information

**S1 File. PROSPERO registration information.**
(PDF)

**S2 File. PRISMA 2020 checklist.**
(DOCX)

**S3 File. Data sources.**
(RM5)

**S4 File. Search Strategy.**
(DOCX)

**S5 File. Quality evaluation of all articles.**
(DOCX)

## Author Contributions

**Data curation:** Yun Li.

**Formal analysis:** Wang Guo.

**Investigation:** Ruo-xia Chen, Xue-rui Lv.

**Methodology:** Wang Guo.

**Project administration:** Yun Li.

**Software:** Ruo-xia Chen, Xue-rui Lv.

**Validation:** Wang Guo.

**Writing – original draft:** Yu-xuan Li.

**Writing – review & editing:** Yu-xuan Li.

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
