## [Decision Letter · Decision Letter 0]

6 Feb 2024

PONE-D-23-34820The relationships between obesity and  Epilepsy:A systematic review with meta-analysisPLOS ONE

Dear Dr. li,

Thank you for submitting your manuscript to PLOS ONE. After careful consideration, we feel that it has merit but does not fully meet PLOS ONE’s publication criteria as it currently stands. Therefore, we invite you to submit a revised version of the manuscript that addresses the points raised during the review process.

**ACADEMIC EDITOR:** - in addition to the comments of the reviewers, please do make following improvements: - formulation of the aim at the end of introduction section should be done better. emphasize that you are performing meta analysis. It's not enough to say that you are investigating "connection"-emphasize limitation of your study in discussion- make a take a home massage based on your results as a concluding paragraph (giving perspectives are fine, but make a clear conclusion). 

We look forward to receiving your revised manuscript.

Kind regards,

Dragan Hrncic

Academic Editor

PLOS ONE

Journal Requirements:

Reviewers' comments:

Reviewer's Responses to Questions

**Comments to the Author**

1. Is the manuscript technically sound, and do the data support the conclusions?

Reviewer #1: Partly

2. Has the statistical analysis been performed appropriately and rigorously? 

Reviewer #1: Yes

3. Have the authors made all data underlying the findings in their manuscript fully available?

Reviewer #1: Yes

4. Is the manuscript presented in an intelligible fashion and written in standard English?

Reviewer #1: Yes

5. Review Comments to the Author

Reviewer #1: Dear Authors

Thank you for this manuscript on the important topic of the relation between obesity annd epilepsy.

The study was done to examine the correlation between obesity and epilepsy hence only case control studies were selected.

I suggest studies of other research design could also be reviewed.

The result should state the difference that were seen when the sensitivity analysis was done for the studies excluding hightest weight.

What was the reason behind the subgroup analysis of years of publication and developed and developing countries?

Also discussion has more part on various drug and their obesity inducing effects which can be tabulated for a better understanding.

It would be more comprehensible if the studies are arranged in chronological order as per the year of publication.

Thank you

Best wishes

6. PLOS authors have the option to publish the peer review history of their article (what does this mean?). If published, this will include your full peer review and any attached files.

Reviewer #1: **Yes: **Harsh Priya

---

## [Author Response · Author response to Decision Letter 0]

8 May 2024

Response to Reviewers

Dear editor, 

first of all,I would like to thank you and the reviewers for your review of our manuscript and your valuable suggestions for revision.We have carefully read your amendment comments and have carefully revised in response to the issues you have raised.We have marked the revised part of the manuscript in red font,and the specific content can be seen in the "Revised Manuscript with Track Changes".Below are our responses to your comments.

 1.Responding to comments from academic editors:

(1)formulation of the aim at the end of introduction section should be done better.emphasize that you are performing meta analysis.It's not enough to say that you are investigating "connection".

Thank you very much for your valuable revision suggestions,so that we can further improve the article.After careful consideration,we have made corrections and additions.We have carefully considered your comments.First of all,we emphasize that we are conducting a meta-analysis,and further explain that the purpose of meta is to better explore the correlation between epilepsy and obesity,and further provide references forclinical diagnosis and treatment of epilepsy.Therefore,our study supplemented the possible influencing factors of obesity and epilepsy,and increased the practical reference value for clinical work.The changes formulated are detailed in the introduction section of the revised manuscript.

(2)emphasize limitation of your study in discussion.

 Our study did have some limitations,and we have added a more detailed supplementary note.See the limitations section of the manuscript for details.

(3)make a take a home massage based on your results as a concluding paragraph (giving perspectives are fine,but make a clear conclusion).

I think you're right.We really don't seem to have a clear conclusion.We have added and modified the conclusions in the results section.See the research results sectionof the manuscript for details.

2.Respond to reviewer's comments:

(1)The study was done to examine the correlation between obesity and epilepsy hence only case control studies were selected. I suggest studies of other research design could also be reviewed. 

In accordance with your suggestions,we have made careful revisions to our manuscript.Your suggestions are very helpful to improve the quality of our manuscripts.On the basis of the original case-control study,we added cohort studies,non-randomized controlled trials and cross-sectional studies,but after further screening,the cohort studies and non-randomized controlled trials were excluded after they failed to meet our screening conditions.After the final screening,we added a cross-sectional study to further expand our research.This makes our research more comprehensive and convincing.

(2)The result should state the difference that were seen when the sensitivity analysis was done for the studies excluding hightest weight. 

We performed a sensitivity analysis by excluding the studies with the highest weight(Fig 5),and the results showed heterogeneity(p=0.008, I2=52%),indicating that the meta-analysis results may be unstable.

(3)What was the reason behind the subgroup analysis of years of publication and developed and developing countries? 

We conducted subgroup analysis of the year of publication and the degree of development of the country in order to find the reasons for the heterogeneity of the meta-analysis results.First,different economic development in different publication years may affect the incidence of obesity and epilepsy.However,subgroups found that publication years were not related to the heterogeneity of meta-analysis results.Secondly,the subgroup analysis of developed and developing countries is also due to the consideration of the impact of economic issues on epilepsy and obesity.However,it is also found that the development level of a country has no significant correlation with the correlation between epilepsy and obesity,and the source of heterogeneity is not found.However,according to your suggestion,we have added some studies and conducted subgroup analysis on the publication years and economic development again,and found that the publication years and economic development level are the sources of heterogeneity.In order to better discuss the source of heterogeneity,we further conducted subgroup analysis and found that age was also the source of heterogeneity in our study.

(4)Also discussion has more part on various drug and their obesity inducing effects which can be tabulated for a better understanding. 

Thank you very much for your advice.We have further increased the impact of various anti-epileptic drugs on obesity,which can indeed better understand and compare the impact of different drugs on obesity,and provide better reference value for clinical drug use.See Table 3 added to the manuscript for details

(5)It would be more comprehensible if the studies are arranged in chronological order as per the year of publication. 

Thank you very much for your suggestion,we have revised it.

Finally,thank you again for your valuable comments and those of the reviewers!At the same time,we hope that if you find any deficiencies again in the process of review,please kindly give us your valuable revision suggestions again,and we will continue to revise carefully.We look forward to publishing our articles on your magazine platform,and we will work harder for this.Looking forward to your good news. 

Yours sincerely,

Yu-xuan Li 

Department of Neurology, The First Affiliated Hospital of Dali University, Dali, China.18408832102 

liyuxuandldx@163.com

---

## [Decision Letter · Decision Letter 1]

12 Jun 2024

The relationships between obesity and  Epilepsy:A systematic review with meta-analysis

PONE-D-23-34820R1

Dear Dr. li,

We’re pleased to inform you that your manuscript has been judged scientifically suitable for publication and will be formally accepted for publication once it meets all outstanding technical requirements.

Kind regards,

Prof.Dr. Dragan Hrncic, MD, PhD

Academic Editor

PLOS ONE

Additional Editor Comments (optional):

Reviewers' comments:

Reviewer's Responses to Questions

**Comments to the Author**

1. If the authors have adequately addressed your comments raised in a previous round of review and you feel that this manuscript is now acceptable for publication, you may indicate that here to bypass the “Comments to the Author” section, enter your conflict of interest statement in the “Confidential to Editor” section, and submit your "Accept" recommendation.

Reviewer #1: All comments have been addressed

2. Is the manuscript technically sound, and do the data support the conclusions?

Reviewer #1: Yes

3. Has the statistical analysis been performed appropriately and rigorously? 

Reviewer #1: Yes

4. Have the authors made all data underlying the findings in their manuscript fully available?

Reviewer #1: Yes

5. Is the manuscript presented in an intelligible fashion and written in standard English?

Reviewer #1: Yes

6. Review Comments to the Author

Reviewer #1: Thank you to the authors. I am convinced with the step by step reply. All the requisite questions are sorted. It may be accepted.

7. PLOS authors have the option to publish the peer review history of their article (what does this mean?). If published, this will include your full peer review and any attached files.

Reviewer #1: No

---

## [Editor Report · Acceptance letter]

28 Jun 2024

PONE-D-23-34820R1 

PLOS ONE

Dear Dr. Li, 

I'm pleased to inform you that your manuscript has been deemed suitable for publication in PLOS ONE. Congratulations! Your manuscript is now being handed over to our production team.

Kind regards, 

on behalf of

Professor Dragan Hrncic 

Academic Editor

PLOS ONE